# Gels in Motion: Recent Advancements in Energy Applications

**DOI:** 10.3390/gels10020122

**Published:** 2024-02-02

**Authors:** Aditya Narayan Singh, Abhishek Meena, Kyung-Wan Nam

**Affiliations:** 1Department of Energy and Materials Engineering, Dongguk University—Seoul, Seoul 04620, Republic of Korea; 2Division of Physics and Semiconductor Science, Dongguk University—Seoul, Seoul 04620, Republic of Korea; pakar.abhishek@gmail.com; 3Center for Next Generation Energy and Electronic Materials, Dongguk University—Seoul, Seoul 04620, Republic of Korea

**Keywords:** gels, fuel cell, Li-ion batteries, supercapacitors

## Abstract

Gels are attracting materials for energy storage technologies. The strategic development of hydrogels with enhanced physicochemical properties, such as superior mechanical strength, flexibility, and charge transport capabilities, introduces novel prospects for advancing next-generation batteries, fuel cells, and supercapacitors. Through a refined comprehension of gelation chemistry, researchers have achieved notable progress in fabricating hydrogels endowed with stimuli-responsive, self-healing, and highly stretchable characteristics. This mini-review delineates the integration of hydrogels into batteries, fuel cells, and supercapacitors, showcasing compelling instances that underscore the versatility of hydrogels, including tailorable architectures, conductive nanostructures, 3D frameworks, and multifunctionalities. The ongoing application of creative and combinatorial approaches in functional hydrogel design is poised to yield materials with immense potential within the domain of energy storage.

## 1. Introduction

Very few things are agreed upon so well in the scientific community as the fact that the global atmosphere will continue to heat up if anthropogenic emissions, particularly CO_2_, are not drastically reduced to net zero [1,2]. Thus, in the pursuit of achieving a carbon-neutral society and attaining net zero emissions, it is mandatory to explore renewable energy sources. The exploration of renewable energy is also essential as fossil fuels have significantly deteriorated the global atmosphere and are yet on the brink of being exhausted in the coming few decades. This alarming scenario forces energy researchers to explore several new renewable sources. However, there is one important property of energy: energy can neither be created nor destroyed; it can just be interconvertible. For instance, photovoltaic cells convert solar radiation to electrical energy [3,4]. For this interconversion, the energy in one form must be stored as chemical energy via a chemical redox reaction [5,6]. The basic idea revolves around thoroughly understanding this redox reaction to scale this to novel materials with a convenient electron transfer so that energy conversion can be efficiently transformed to electrical energy and stored for further use. It is this interconversion reaction and the electrochemical reaction that are swiftly attracting attention from the research community to alleviate the global warming issue. This review underscores the use of gels in different energy conversion and storage reactions.

Gels are a special class of semi-solid materials with an intermediate consistency between solid and liquid state [7]. Out of different types of gels, polymeric gels have emerged as promising materials in the realm of energy applications, showcasing their versatility and unique properties that make them invaluable in various fields. These gels, composed of long chains of polymers dispersed in a liquid phase, exhibit a distinctive ability to undergo reversible changes in their physical state, transitioning between a solid and liquid form. This remarkable characteristic has sparked significant interest in harnessing polymeric gels for energy-related endeavors. In the context of energy applications, polymeric gels offer a wide range of functionalities, from serving as electrolytes in advanced batteries to playing a crucial role in developing smart materials for energy storage and harvesting. Their tunable properties, such as mechanical strength, thermal conductivity, and responsiveness to external stimuli, make them ideal candidates for creating efficient and adaptable systems. This introduction will delve into the key attributes of polymeric gels that make them well-suited for energy applications, exploring their potential impact on advancing renewable energy technologies, energy storage devices, and other innovative solutions. As researchers continue to unlock the full potential of polymeric gels, the future holds exciting possibilities for these materials in shaping a sustainable energy landscape.

Polymeric gels, formed by creating cross-links between polymers in a solution phase, are a key material in various manufacturing applications. Cross-linked polymers typically exhibit increased resilience under mechanical deformations and can absorb significant amounts of solvents, particularly when there is a high affinity between polymers and solvents. Adding filler particles and other suitable particles inside the polymer gel matrix can easily tailor the properties of these materials. However, when it comes to application in the fields of energy materials, conducting polymers plays a vital role. Polymeric gels, particularly conducting polymer gels, have added benefits of excellent flexibility, large surface area, three-dimensional framework, high mechanical strength, elasticity, optoelectronics, electrochemical properties, etc. The content of the entire review is pictorially represented in Figure 1.

### 1.1. Hydrogels

Hydrogels hold an important place in the gel category due to their wider applicability in various domains, not just limited to energy, and thus, we dedicate a separate section to this. Hydrogels are a special class of 3D polymeric network chains capable of withstanding large volumes of water while being water-insoluble [8]. It has been a hot area of research, particularly due to tunable physicochemical properties such as pore size, stiffness, microarchitecture, etc. The ability of the hydrogels arises due to hydrophilic functional groups attached to the backbone of the polymers, while their water-insoluble characteristics evolve from cross-linking between the network chains. While this section and the review primarily concentrate on applying hydrogels and elucidating the underlying chemistries, we categorize hydrogels into natural and synthetic types to simplify. However, synthetic hydrogels have replaced natural hydrogels over time due to unique properties like high gel strength, better ability to hold water, and long cycle life. Synthetic polymers also offer a tailor-made capability to design a required material of choice with better degradability and functionality. A few other important classifications are also discussed in the literature depending upon their physical and chemical compositions, which are (i) amorphous (non-crystalline), (ii) crystalline, and (iii) semicrystalline. A few studies also classify them based on the presence of electrical charges on the polymeric backbone. In this way, the first could be ionic type, the second non-ionic (neutral), the third comprises ampholytic (containing both acidic and basic groups), and the fourth class is Zwitterionic [9]. For the general description of the hydrogels classification based on their sources, Figure 1 provides a reference.

#### Stimuli-Responsive Polymer Hydrogels

Stimuli-responsive hydrogels, as the name suggests, are gels that are sensitive to the changes in the external environment and act accordingly by bringing a change in their structural or mechanical changes [10]. This new polymeric hydrogel class is increasingly gaining attention in medical and robotics applications due to its ability to respond to changes in external environments. Various kinds of stimuli are light/photon [11], magnetic field [12], electric field [13], temperature [14], biological triggers [15], pH [16], and some chemicals can be used to bring desired phase changes inside hydrogels to manifest different applications such targeted drug delivery [17], sensing, bionic devices [18], regenerative medicines, and more (Figure 2) [19,20].

Before discussing the application areas, the mechanism behind stimuli-responsive hydrogels must be understood. The formation of hydrogels is normally completed in three steps, as discussed and presented first in 2010 [21]. Firstly, the hydrophilic groups of gelators interact with water molecules, forming primary-bound water. Subsequently, water molecules engaged with exposed hydrophobic groups, constituting the secondary bound water. In the third stage, the osmotic force generated by the gelators attracted a significant influx of water molecules into the network, initiating the dissolution process. This dissolution was counteracted by the presence of cross-links, resisting the complete breakdown of the system. Eventually, a state of equilibrium was achieved, resulting in the formation of a stable hydrogel.

Hydrogel possesses a unique self-healing property that makes it the most researched material suitable for cell repairing for biomedical applications [22]. The mechanistic origin of self-healing emanates from the reversibility of the cross-linking structures present in the backbone of the hydrogels. This self-healing property of the hydrogels has made them a versatile biomaterial due to their excellent ability to repair the initial structure in response to the damaged cell. Several characterization methods exist to determine or detect hydrogels’ self-healing properties, including spectroscopic and X-ray techniques. Along with these, there are also some less commonly used methods like thermal analysis. These techniques confirm the successful generation of hydrogels in experiments and the intended creation of reversible interactions. However, for intricate structures, relying on a single method might be insufficient to fully characterize the structure, necessitating combining different techniques.

A recent study revealed that tailor-made hydrogels can be designed as excellent sensors to respond to varying temperatures due to their dynamic cross-linking structures [23]. This work addresses the limitations of traditional hydrogel sensors, which can freeze or evaporate under extreme conditions and have low strength and high resistance, limiting their practical applications. This work presents a novel technique involving the covalent interaction of acidified carbon fibers and various ions to create mechanically strong and conductive nanocomposite hydrogels. The resulting hydrogel, comprising polyacrylic acid, sodium alginate, and acidified carbon fibers, exhibits high sensitivity, flexibility, self-healing, anti-freezing, and anti-drying properties. The interpenetrating cross-linked network structure ensures excellent ductility, repeatability, and stable sensing performance. The hydrogel strain sensor, suitable for real-time human movement detection, demonstrates a low strain detection limit, fast response time, high sensitivity, and self-adhesion to various substrates. It maintains superior flexibility, extensibility, and conductivity across a wide temperature range, from −20 °C to 50 °C, enhancing its practicality and durability. This hydrogel is promising for wearable devices, soft robot systems, and health monitoring applications. In another study, a near-infrared (NIR)-induced rapid self-healing hydrogel was designed to demonstrate multifunctional application in sensing [24]. Here, in the design of self-healing hydrogels, the authors claim that they have used inorganic antimony sulfide (Sb_2_S_3_) as a photothermal agent for efficient self-healing sensors for the first time. A rapid self-healing double-network conductive hydrogel (SPOH gel) was created using near-infrared (NIR) light-induced on-demand irradiation. The gel, incorporating dopamine (DA)-modified polypyrrole-coated Sb_2_S_3_ nanorods (Sb_2_S_3_ @PPy-DA) into a polyvinyl alcohol (PVA)/poly(N-(2-hydroxyethyl) acrylamide) (pHEAA) matrix, displayed adhesive and anti-freezing properties and robust mechanical characteristics (1.25 MPa tensile strength, 620% elongation, 251 J m^−2^ interfacial toughness, GF = 4.97 sensitivity). The inclusion of Sb_2_S_3_ @PPy-DA resulted in high conductivity (2.26 S/m) and facilitated rapid self-healing under NIR light (within 90 s). The SPOH gel demonstrated versatility in applications, serving as a strain sensor for monitoring human motion, biopotential electrodes for electrocardiogram signal detection, and a self-powered device for energy harvesting. This multifunctional SPOH gel, with its rapid self-repair and real-time physiological signal detection, offers promising prospects for wearable smart devices in motion monitoring, healthcare, and energy harvesting applications.

### 1.2. Physical Properties

Gel is a thing that is easier to recognize than assign a proper definition. According to P.J. Flory, the most holistic definition of gel stands out as a continuous structure that is permanent on the analytical time scale while it is solid-like in rheological behavior [25]. There are peculiar features of gels that set them apart from other materials. (i) Gels are reversibly compressible, which means they can undergo deformation (compressed) under applied stress, and when the stress is removed, gels can revert to their original state, demonstrating reversibility. This behavior is due to the unique structure of gels, which typically consist of a three-dimensional network of polymers or particles embedded in a liquid medium. The compressibility of gels is an important property in various applications, including in the pharmaceutical, food, and cosmetic industries [26]. (ii) The fibrillar network framework structure achieved by the cononsolvency effect [27] is another salient feature of gel polymers that allows them to self-organize the complex systems in living organisms [28]. (iii) Storage modulus (G′) and loss modulus (G″) [29] are other crucial aspects that determine the measure for the fraction of deformation energy induced by motor movement and elastically retained in the sample. This measurement captures insights into the internal structure of the three-dimensional network within the gel system [30]. The distinctive solid-like behavior of a gel originates from its inherent 3D framework that encapsulates the liquid from which the gel is derived. This encapsulation prevents the flow of the liquid, resulting in the semi-solid nature of the gel. Gels, by property, are semi-solid in nature and thus behave intermediately between solid and liquid. Gels are primarily produced by the interactions of physical (non-covalent) or chemical (covalent) and sometimes by cross-linking agents [31], which are also involved between their building blocks. Gels formed by covalent bonding/interactions are called chemical irreversible gels, while gels formed by physical interactions such as hydrogen bonding [32], Coulombic interactions [33], π-stacking [34], or hydrophobic interactions [35] are often termed as thermos-reversible gels [36]. In the polymer gel, the building blocks may be natural polymers like gelatin, agarose, chitosan, or synthetic non-conducting polymers like polyvinyl alcohol (PVA), polyacrylic acid (PAA), and polyethylene glycol (PEG), or sometimes can be synthetic conducting polymers [37] such as polypyrrole, polyaniline, or polythiophene. These building blocks play a crucial role in determining the properties of gels, and when it comes to their application in energy storage/conversion devices, the selection of these building blocks becomes even more critical. When applied to storage/conversion devices, conducting polymeric gels are mostly used as conjugated chains sensitive to external factors such as stress, temperature variations, pH variations, voltage, and light. Apart from the conducting polymer gels, ionic conducting gels are also essential during device fabrications as the mobility of ions plays a significant role in conductivity [38].

### 1.3. Gel Mechanics

Since the gel is a composite structure consisting of solid and liquid, the mechanical response of gels is a combined response of its constituent phases and sometimes to their interactions as well [39]. In particular, when a gel is subjected to an applied load, a pressure gradient is built up, which causes the flow of liquid; consequently, the gel bears a time-dependent response to externally applied loads, even when the gel is not supposed to be a viscoelastic material by itself. However, in cross-linked chemical gels, due to strong cross-links, these gels are resistant to flow or melt or do not have a viscoelastic state at all. So, depending upon the types of gels, their physical response can vary significantly. Sometimes, the cross-linked polymeric gels become strictly reluctant and unable to move past each other or around each other, and this possibly causes many of these polymeric gels to not being soluble in several solvents. Indeed, to become soluble, this cross-linking must be loosened enough to move apart to spread in the entire volume of the solvent and become soluble; however, this is daunting in the presence of strong cross-linking.

## 2. Energy Applications

This section deals with the extensive applications of gels in batteries, fuel cells, solar cells, supercapacitors, and other energy applications. Due to their unique properties, polymeric gels play a pivotal role in various energy applications. They serve as electrolytes in battery technology, offering high ionic conductivity and improved safety. Additionally, polymeric gels are found to be used in solar cells, enhancing efficiency and durability. Their tunable optical properties make them valuable for smart windows, contributing to energy-efficient building designs. Fuel cells benefit from polymeric gels as electrolytes or membrane components, ensuring optimal performance. These gels also play a role in thermal energy storage oil recovery processes and are key elements in energy harvesting devices. The versatility of polymeric gels underscores their significance in advancing technologies for sustainable energy solutions.

### 2.1. Supercapacitors

Supercapacitors are one of the basic modes for electrochemical energy storage (EES) other than batteries. Here, energy storage occurs via ion storage on the electrode surface. Supercapacitors store electrical energy through a dual mechanism involving double-layer capacitance and pseudocapacitance. In the double-layer capacitance mechanism, charges accumulate at the interface between the electrode material and the electrolyte, forming an electric double layer (EDL). This electrostatic separation of charges results in a highly reversible process, allowing for swift charge and discharge cycles. Simultaneously, some supercapacitors leverage pseudocapacitance, which involves redox reactions within the electrode material, typically using substances like transition metal oxides or conducting polymers. This Faradaic process contributes to a higher energy density than pure double-layer capacitance, albeit with a slightly slower response. The synergy of these mechanisms enables supercapacitors to combine the rapid charge/discharge capabilities of capacitors with the higher energy storage capacity typical of batteries, making them valuable in applications requiring quick bursts of energy and frequent cycling, such as regenerative braking in electric vehicles and energy storage in renewable systems [40,41].

#### 2.1.1. Hybrid Polymer Gel Electrodes for Supercapacitors

Polymer gel electrodes represent a specialized class of electrodes with distinctive properties that set them apart in various applications. Comprising a gel-like matrix, these electrodes exhibit unique characteristics such as high flexibility, mechanical stability, and enhanced ion conductivity. The gel matrix, often formed by polymers or hydrogels, serves as an efficient medium for ion transport and accommodates volumetric changes during electrochemical processes. This feature is particularly advantageous in applications demanding flexibility and adaptability, such as wearable devices or bioelectrochemical systems. The gel-type classification is attributed to the presence of this flexible matrix, not only providing mechanical support but also facilitating improved contact between the electrode material and the electrolyte. Understanding these properties and their classification as gel types is crucial for appreciating the distinct advantages polymer gel electrodes offer in various electrochemical and biomedical applications.

Alshareef and co-workers [42] reported a detailed study on developing solid-state asymmetric supercapacitors (ASCs) using conducting polymer electrodes on a flexible plastic substrate. Electrochemically deposited nanostructured conducting polymers, poly(3,4-ethylenedioxythiophene) (PEDOT) and polyaniline (PANI), are utilized on Au-coated polyethylene naphthalate (PEN) plastic substrates. With its electron-donating oxygen groups, PEDOT serves as a negative electrode due to its higher reduction potentials. Its stability in the oxidized state allows it to exhibit electrochemical activity across a wide potential window. The ASCs, employing PANI as a positive electrode and polyvinyl alcohol (PVA)/H_2_SO_4_ gel electrolyte, demonstrate impressive performance with a maximum power density of 2.8 W cm^−3^ at an energy density of 9 mW h cm^−3^. This surpasses carbonaceous and metal oxide-based ASC solid-state devices. Additionally, the tandem configuration of these ASCs successfully powers a red light-emitting diode for approximately 1 min after a 10 s charge, showcasing their potential for practical applications. A recent study by Lee’s group [43] presented a novel approach to fabricating a symmetric supercapacitor with multiple functionalities, including high capacitance, high-rate capability, flexibility, and self-healing properties (Figure 3a). The design involves a composite current collector electrode comprised of a self-healable material (poly(3,4-ethylenedioxythiophene): poly(styrene sulfonate)/multiwalled carbon nanotubes) and a poly(vinyl alcohol)/phosphoric acid electrolyte. The key innovation lies in the use of a hybrid carbon nanofiller (HCF) coupled with healable polymer matrices (HCF/HPU), forming a self-healing, highly conductive composite current collector. The self-healing property was demonstrated by the recovery of capacitance in successive cycles (Figure 3b). The Nyquist plot indicates that its resistance increases slightly over their continued cycles (Figure 3c). This composite exhibits enhanced electrical properties and mechanical strength compared to single nanofillers. The resulting supercapacitor demonstrates high specific capacitance, energy density, and cycling stability, with only ~10% capacitance loss after 20,000 charge/discharge cycles at a high current density of 10 mA cm^−2^. Moreover, the HCF/HPU composite current collector supercapacitors exhibit remarkable self-healing capabilities, retaining over 96.2% of their capacitance after severing/healing cycles and maintaining 97.4% capacitive retention after 2000 bending cycles (Figure 3d). These findings position the HCF/HPU composite current collector as a promising candidate for high-performance flexible energy storage devices. In another study on electrodes to potentially improve the supercapacitor performances, nanoscale α-NiMoO_4_ particles encapsulated within electronically conducting polymer nanocomposites (PNCs) were developed using a base of Polyvinyl alcohol (PVA)/Poly(vinyl) pyrrolidone (PVP) [44]. The electrochemical characteristics of the developed polymer nanocomposites (PNCs) were examined, revealing a maximum specific capacitance of 15.56 F g^−1^ for a polymer-blended electrode loaded with 1 wt % of α-NiMoO_4_ nanoparticles (NPs) at a scan rate of 5 mV s^−1^. When evaluated in a two-electrode system at room temperature, the PNCs displayed an impressive 97.12% Coulombic efficiency at room temperature, using a 3 M KOH aqueous electrolyte solution. The CV curve shows a larger scan area at a given potential for the synthesized electrode (Figure 3e), while the specific capacitance at a varying scan rate is plotted in (Figure 3f). It is revealed that the synthesized electrode becomes crystalline due to the increase in cross-linking densities of the composites, leading to high capacitance retention, as shown in (Figure 3g). These hybrid composite gel combustion synthesis techniques can significantly improve the electrochemical performance of the supercapacitors.

#### 2.1.2. Hybrid Polymer Gel Electrolytes for Supercapacitors

As far as the application of gel polymers as electrolytes in supercapacitors is concerned, there are several advantages over other classes of electrolytes [45]. With the rapid growth in portable electronic devices, there is an urgent need to develop wearable, flexible energy storage devices like flexible supercapacitors (FSCs) [46,47,48,49,50]. To achieve high performances out of supercapacitors, the development of better electrodes, housing, separators, and other flexible components are crucial; however, this section is dedicated to the development of gel electrolytes to enhance the electrochemical performances of supercapacitors. Hybrid polymer gel electrolytes represent an innovative approach in advancing the performance of supercapacitors. In these systems, traditional liquid electrolytes are combined with polymer gels, resulting in a hybrid material that harnesses the benefits of both components. The polymer gel provides mechanical stability and flexibility and prevents electrolyte leakage, addressing common challenges associated with liquid electrolytes. This combination enhances the overall safety and structural integrity of supercapacitors. The polymer matrix can also contribute to improved ion conductivity and charge/discharge efficiency. The synergy between the liquid electrolyte and the polymer gel in hybrid systems often leads to enhanced electrochemical performance, making them promising candidates for next-generation supercapacitor technologies. Researchers are actively exploring various polymer gel formulations to optimize conductivity, mechanical properties, and overall energy storage capabilities in the quest for efficient and reliable energy storage solutions.

However, before moving to explore the developments in the field of gel electrolytes, it is crucial to understand how FSCs work. FSCs can be categorized into two distinct types: (i) electric double-layer (EDL) capacitors and (ii) pseudocapacitors. In the case of EDL capacitors, energy is stored and released through the adsorption/desorption process of electrolyte ions on the surface of the electrode material. In contrast, pseudocapacitors, also known as Faraday quasi-capacitors, operate on a different energy storage mechanism. They involve a rapid and reversible series of redox reactions with electrolyte ions, occurring either at the surface or within the bulk phase of the electrode material. In 1853, EDL capacitance was first proposed by the German physicist Helmholtz [51]. As shown in Figure 4a, the generation of EDL capacitance results from the alignment of charges arising from the directional arrangement of electrons or ions at the electrode interface and solution [52,53]. In an electrode/solution system, applying an electric field to both electrodes induces the migration of solvated anions and cations to the positive and negative electrodes, respectively. This process gives rise to the formation of a Helmholtz double layer with a 1 nm thickness at the electrode surface. Upon removing the electric field, positive and negative ions are released from the electrode surface into the electrolyte. The stability of the electrical double-layer is maintained through the attraction of opposite charges, leading to the establishment of a relatively stable potential difference between the positive and negative electrodes. In a similar way, pseudocapacitors produce capacitance associated with the charging potential of the electrode in a two-dimensional or quasi-two-dimensional region on the electrode surface and in proximity to the surface or within the bulk phase (Figure 4b). This is attributed to underpotential deposition or a highly reversible chemical adsorption/desorption or oxidation/reduction reaction of the electroactive material. In the context of a Faraday quasi-capacitor, the charge storage process involves the electric double layer storage and redox reactions between electrolyte ions and the active electrode materials.

The use of flexible polymer gel electrolytes in FSCs is gaining much attention to improve the electrochemical performances of supercapacitors. In this direction, several researchers have developed several gel electrolytes, such as PVA/H_3_PO_4_ [55], PVA/H_2_SO_4_ [56,57,58,59], PVA/KOH [60,61,62], PVA/LiOH [63], PVA/Na_2_SO_4_ [64], PVA/KNO_3_ [65], PVA/KCl [66], PVA/LiCl [67], etc., to enhance the versatility of gel electrolytes for use in various conditions. Esawy et al. [68] extensively studied different aqueous electrolytes such as KOH, H_2_SO_4_, H_3_PO_4_, and six different gel electrolytes (PVA/KCL, PVA/H_3_PO_4_, PVA/H_2_SO_4_, PVA/KOH, PVA/KOH–KCl–K_3_[Fe(CN)_6_], and PVA/KNO_3_) and employed them in the fabrication of flexible supercapacitors. The electrochemical characteristics of these diverse electrolytes are assessed through cyclic voltammetry, galvanostatic charge/discharge curves, and impedance spectroscopy. Among these, the capacitor utilizing a PVA–KOH–KCl–K_3_[Fe(CN)_6_] electrolyte membrane with a weight ratio of 60:23:23:4 exhibits the highest specific capacitance of 520 F g^−1^. Furthermore, it demonstrates a prolonged cycling life, with 98.1% retention after 1000 cycles, and its specific capacitance shows an increase with temperature elevation from 25 to 70 °C. The enhanced electrochemical performances of this gel electrolyte is due to its larger electrode capacitance (520 F g^−1^) and reduced charge-transfer resistance (R_ct_) value. To further improve the electrochemical performances of supercapacitors, gel electrolyte modification on D-A-D-type conjugated polymer technique was adapted [69]. To significantly enhance the device performance, TiO_2_ NPs were integrated into the PMMA/TBAPF_6_-containing gel electrolyte, with the optimization of the TiO_2_. An irregular porous structure was revealed under SEM (Figure 5a). The impact of TiO_2_ addition on device performance was evident in the increased ionic conductivity observed. The highest area-specific capacitance was recorded at 3.30 mF cm^−2^ at 2 mV s^−1^ and 3.08 mF cm^−2^ at 0.015 mF cm^−2^. The plots of the CV curves obtained shows onset oxidation potential at 0.75 V, much lower than EDOT, resulting from enlarged conjugated block (Figure 5b). A plot of specific capacitance and its peculiar decline on increased current density is shown in Figure 5c. The maximum energy density reached 1.47 μW·h cm^−2^, accompanied by a corresponding power density of 13.5 μW cm^−2^. Furthermore, the electrochemical stability of the supercapacitor devices was assessed through 1000 galvanostatic charge–discharge (GCD) cycles, revealing excellent stability in terms of both % capacity retention and % Coulombic efficiency. The improved performance of this device can also be understood by a reduced R_ct_ value, shown in Figure 5d. Another study showed that supercapacitor’s performances can be further improved by using liquid crystal gel electrolyte [70]. This study developed a sunlight-powered energy storage system using single-wall carbon nanotubes (SWCNTs)-based symmetric micro-supercapacitors (MSCs) fabricated through a facile one-step spraying method. The electrodes exhibited satisfactory conductivity, and all-solid-state MSC devices were designed using potassium hydroxide-poly(vinyl alcohol) (KOH-PVA) and phosphoric acid-nonionic surfactant liquid crystal (PANI LC) gel electrolytes. The SWCNTs/PANI LC electrolyte demonstrated superior performance, exhibiting a larger areal capacitance of 11.0 mF·cm^−2^ at room temperature. The CV and GCD measurements are shown in Figure 5e,f. Temperature effects on device performance were investigated, revealing enhanced electrochemical performance at 65 °C, including a larger areal capacitance of 14.7 mF·cm^−2^, better rate performance, and higher energy density. The MSC device also demonstrated high cyclic stability under bending conditions. Moreover, a sunlight-powered energy storage system was constructed, combining solar cells with the MSC devices, showcasing continuous operation of a red light-emitting diode for 2 min and 30 s (Figure 5g–i). These results suggest the promising practical application potential of the MSC devices as efficient energy storage solutions, particularly at elevated temperatures.

## 3. Fuel Cells

Fuel cells and gel-based technologies represent innovative and sustainable solutions in the realm of energy conversion and storage. Fuel cells are electrochemical devices that convert chemical energy directly into electrical energy through the reaction between fuel and an oxidizing agent, typically oxygen or air. This process offers high efficiency and low environmental impact, making fuel cells a promising alternative to traditional combustion-based power sources. On the other hand, gel-based technologies, particularly gel electrolytes, have gained prominence in various energy storage applications. Gel electrolytes provide a solid-like consistency, offering advantages such as improved safety, flexibility, and ease of handling compared to traditional liquid electrolytes. They find application in diverse energy storage devices, including supercapacitors and batteries, enhancing their performance and enabling novel designs for portable electronics, electric vehicles, and renewable energy systems. The intersection of fuel cells and gel technologies presents an exciting frontier in pursuing efficient and sustainable energy solutions. Researchers are exploring the integration of gel electrolytes in fuel cells to address challenges related to liquid electrolyte leakage and improve overall device performance. This synergy holds the potential to advance the development of reliable and versatile energy conversion and storage systems, contributing to a cleaner and more sustainable energy landscape [71].

In the study of Wang et al. [72], aq. H_2_SO_4_ solutions with concentrations ranging from 1 to 4 mol L^−1^ are fabricated into gel membranes through the in situ polymerization of acrylamide as a monomer and divinylbenzene as a crosslinker (Figure 6a). The obtained mixture was poured into a silica mold with a piece of a glass tube and irradiated under UV (365 nm) to turn it into a transparent gel (Figure 6b). This process is advantageous for electrochemical applications and offers benefits such as easy shaping and reduced leaking. The gel membrane with a sulfuric acid concentration of 3.5 mol L^−1^ demonstrates a peak proton conductivity of 184 mS cm^−1^ at 30 °C. Additionally, the tensile fracture strength of the gel membrane reaches 53 kPa with a tensile strain of 14. Thermogravimetric analysis indicates thermal stability up to 231 °C. Furthermore, the gel membranes are effectively integrated into fuel cells (Figure 6c), achieving a peak power density of 74 mW cm^−2^ (Figure 6d). Notably, the fuel cell maintains steady operation for over 200 h. The in situ gelation of aqueous sulfuric acid solutions presents an efficient strategy for preparing gel electrolytes tailored for electrochemical devices, showcasing promising results for fuel cell applications.

## 4. Battery Technology

Gel electrolytes have emerged as a valuable component in battery technology [73,74,75], addressing key challenges and enhancing overall performance. These semi-solid or gel-like materials, often employed as alternatives to liquid electrolytes in lithium-ion batteries, contribute to improved safety by minimizing the risk of leakage and enhancing electrolyte stability. Notably, gel electrolytes play a crucial role in preventing the formation of dendrites on battery electrodes, reducing the potential for short circuits and enhancing the longevity of the battery. Their flexible form factors allow for molding into various shapes, accommodating diverse battery designs. Furthermore, gel electrolytes exhibit high ionic conductivity while providing mechanical stability, ensuring efficient ion transport within the battery. In the modern day, the concept of fiber batteries is emerging, and we might expect better research results that may improve the intrinsic limitations in the current used materials, such as limited flexibility, which causes internal injuries and physical irritation during contact with biological cells. Therefore, it is imperative to design biologically compatible and ultrasoft materials for flexible batteries. In this particular application, the use of hydrogels can be a game changer [76]. Apart from that, the use of gels is seemingly increasing in various energy storage applications. With applications spanning portable electronic devices to electric vehicles, gel electrolytes continue to be a focal point in battery research, where ongoing efforts aim to optimize their composition for enhanced performance, safety, and versatility in different environmental conditions.

Battery technologies, in particular sodium-ion and lithium-ion batteries (SIB/LIB), often suffer from several issues [6,77,78]. Some of the issues are synthesis-induced and some other evolve during cycling of the cell [79,80,81,82]. In all such cases, gel can play a dominant role in overcoming those issues. Thus, it is imperative to explore gels and its variants in resolving those defects in batteries to improve their electrochemical performances.

### 4.1. Gel Electrolyte Membrane in LIB Technology

Gel electrolyte membranes represent a significant advancement in lithium-ion battery (LIB) technology, offering a solution to safety and performance challenges. These semi-solid or gel-like materials provide enhanced stability for ion transport, effectively addressing concerns related to dendrite formation and electrolyte leakage in LIBs. Serving as a protective barrier between electrodes, the gel membrane prevents short circuits and contributes to the overall structural integrity of the battery. By inhibiting dendrite growth, a common issue in traditional batteries, gel electrolytes significantly enhance the safety and longevity of LIBs. Additionally, the gel format enables greater design flexibility and integration into various form factors while maintaining the high ionic conductivity crucial for efficient lithium-ion movement within the battery. Ongoing research aims to optimize gel electrolyte formulations, promising safer, more efficient, and adaptable lithium-ion batteries suitable for a wide range of applications, from portable electronics to electric vehicles. Li et al. [83] fabricated hybrid PVDF/PEO nanofibrous membranes through the electro-spinning of a PVDF/PEO solution in DMF at various weight ratios (2:1, 5:1, and 10:1), denoted as F/O-2, F/O-5, and F/O-10, respectively (Figure 7a). These membranes, integral components of hybrid gel polymer electrolytes (HPGEs), underwent activation within a glove box. The activation process involved immersing the membranes in a liquid electrolyte (1 M LiPF6 in propylene carbonate (PC)/ethylene carbonate (EC)/dimethyl carbonate (DMC) in a 1:1:1 volume ratio) for one hour. Subsequently, the activated membranes were dried by gently wiping them with filter paper. This detailed procedure outlines the synthesis and activation steps involved in creating the PVDF/PEO hybrid membranes and their integration into HPGEs, providing clarity on the experimental methodology. In another study, a novel approach involving a combination of physical blending and chemical cross-linking procedures is employed to fabricate gel polymer electrolyte membranes with exceptional thermal stability [84]. Initially, precursor porous membranes are crafted using a phase-inversion technique, utilizing poly (vinylidene fluoride) (PVDF) and polystyrene–poly(ethylene oxide)–polystyrene (PS–PEO–PS) triblock copolymer composites. The subsequent step involves an in situ hyper cross-linking process specifically targeting the PS segments in the precursor membranes. This cross-linking procedure effectively consolidates the pore structure, significantly enhancing the thermal stability of the resulting cross-linked composite membranes. Membranes featuring optimal PS/PEO ratios exhibit sustained porosity with minimal dimensional shrinkage even at elevated temperatures, reaching up to 260 °C. The gel polymer electrolytes derived from these cross-linked membranes demonstrate substantially higher ionic conductivities (reaching up to 1.38 × 10^−3^ S cm^−1^ at room temperature) compared to those based on pure PVDF membranes. Assembled Li/LiFePO_4_ half cells with these gel polymer electrolytes showcase commendable cycling performance and rate capability (Figure 7b,c). These findings underscore the efficacy of Friedel–Crafts reaction-based hyper cross-linking as a potent method for constructing polymer electrolytes with remarkable heat resistance, especially beneficial for applications requiring elevated-temperature conditions in lithium-ion batteries. In another study, it was revealed that a gel that is incorporated directly into the commercial polyethylene (PE) separator through simultaneous electron-beam irradiation cross-linking of the conventional liquid electrolyte and poly(ethylene glycol) methacrylate (PEGMA) oligomers can significantly enhance the performance of LIB [85]. These stable cross-linked gel polymers reinforced membranes (GPRMs) offers exceptional mechanical stability and high ionic conductivity compared to PE. This high conductivity is due to the lower resistance of the synthesized material.

The inclusion of hydrogels in battery technology, mainly in the design of fiber batteries, has attracted substantial attention from the research community. Including hydrogels mainly to design biocompatible batteries with the capacity to lower the Young’s modulus of ~450 kPa is a new era of research [86]. The synthesized material with polyacrylamide (PAM)/CNT hydrogel fiber and PAM/CNT hydrogel film demonstrated a high specific discharge capacity of 84.8 mAh·g^−1^ @ 0.5 A·g^−1^. The superior rate and cycling stability is due to better accommodation of lattice strains in hydrogel fibers. The lower Young’s modulus of the material allowed better compatibility with the biological tissues, further precluding skin irritation. Crosslinked hydrogels are another material gaining popularity for their application in fully stretchable solid-state LIB [87]. The authors synthesized all fully stretchable solid-state LIB (FSSLIB) by assembling aneroid components, including a crumpled-structured nanowire (NW) current collector, anode/cathode, and crosslinked hydrogel electrolyte. The wrinkled NWs, interconnected with active material islands, and robust electrode interfaces from hydrogel electrolytes contribute to the electrochemical stability during extensive stretching. The FSSLIB achieves 100% stretchability, displaying a specific capacity of 119 mA h g^−1^ and retaining 91.6% capacity after 250 cycles. This design enhances tensile strength and storage capacity by leveraging the mechanical and electrical properties of NWs and hydrogels, presenting potential applications in large-strain wearable and implantable energy electronics. Hydrogels are not only limited to LIB, but are also applicable to other alkali metal batteries. The use of ionically conducting hydrogels as an electrolyte due to high water in addition to hierarchical pores enables mixed ionic and electronic conductors (MIECs) [76]. In addition to high ionic conductivity (1.76 × 10^−2^ S cm^−1^) [88], hydrogels have high flexibility and mechanical tuneability that can improve the performance of energy storage devices. In another design approach, fine-tuning hydrogel water stages leads to significant improvements in aqueous batteries [89]. Here, the authors investigated water stages in hydrogels extensively and found that by precisely tuning supramolecular hydrogels, it is easy to obtain non-freezable water stages or much less free water stages than found in conventional hydrogels. This technique could extend the voltage window to ~3.25 V with much higher ionic conductivity than found in aqueous K-ion battery. The enhanced performances in the battery is due to the presence of large bound water that can isolate free water from the electrode surface and can restrict side reactions.

### 4.2. Challenges of Using Hydrogels

Despite the numerous benefits of hydrogels, for instance, their mechanical tuneability, flexibility, and lower Young’s modulus for designing biological competitiveness, certain issues must be addressed. The extent of the research activities is quite satisfying, and it is believed that many of the issues discussed could be addressed in the near future, but several brainstorming challenges still persist, which are discussed below in detail.

#### 4.2.1. Hydrogels and Global Energy Consumption

Adopting hydrogels in modern energy consumption and looking at the benefits of hydrogel is a win-win situation. However, there are potential benefits and risks, which will be discussed hereafter.

The use of hydrogels in the energy sector can offer an alternative to store excess produced energy and use it during peak hours. Due to better energy efficiency, the use of hydrogels can enhance the energy efficiency per cycle due to their faster charge–discharge cycles leading to high energy density compared to conventional materials [90]. Novel energy conversion technologies are critical in mitigating greenhouse gas emissions [91,92]. The increasing demand for self-powered energy sources driven by wearable devices has driven energy harvesting to utilize the surroundings and the human body as their source. This has opened a new horizon for the growth of hydrogels due to their 3D-nanostructured polymeric network to set better charge carrier pathways and offer other benefits like tunable mechanical properties. Doping is another technique that can enhance the charge carrier capability of hydrogels and assist in offering larger active sites often required for various electrochemical energy conversion reactions. In addition, their biocompatibility is an added advantage for their rapid progress in applying energy materials [93]. Furthermore, the ionic thermoelectric conductivity in hydrogels can be further enhanced by understanding the Soret effect [94]. Under this phenomenon, a temperature gradient causes a gradient flow of fluid molecules in the hot zones to diffuse along the cold zone. A greater understanding of this phenomenon is essential to better apply this concept in other energy applications. Hydrogels are now used as potential carbon dioxide adsorbents by synthesizing polyacrylic acid (AAc), sodium polyacrylate (SAP), and polyacrylamide (AAm) via solution polymerization technique infused with monoethanolamine (MEA) and diethanolamine (DEA) [95]. The capture of CO_2_ in the porous structure of hydrogels is an effective approach to reduce free carbon in the environment and reduce greenhouse gas emissions [96].

##### Critical Challenges

Despite several benefits, there are some critical issues in the use of hydrogels. Some hydrogel materials may rely on resources that are not abundant, and their extraction could have environmental consequences. Ensuring sustainable material sourcing is essential. While hydrogels function better in aqueous electrolytes, the electrochemical reactions in several energy conversion reactions are limited to constrained voltage windows. Furthermore, water molecules are restricted in a wide range of temperatures. Other issues, such as enhanced understanding of surface chemistry, are vital to tuning charge transfer/transport reaction mechanisms across the various electrode/electrolyte interfaces [97]. Understanding the gelation process is another issue that must be thoroughly understood for the broad applicability of hydrogels in the energy sectors [76].

#### 4.2.2. Production Challenge

Despite several benefits, upscaling is one of the most intricate challenges in hydrogels. The induced bonding agent during polymerization is critical during polymerization in hydrogels and thus is a determining step. The major production challenges come from hydrogel application in the biomedical field. 3D-printed hydrogels used for wound healing and regenerative medicine have yet to implement strict regulations [98]. The absence of strict or uniform regulation imposes a major challenge in techniques like 3D-bioprinting, which is added with high cost and imposes enormous production challenges, particularly in intercontinental transport [99]. In the production of bioink formulation with the capabilities of cell viability and printability, the rate of curing is an essential step that determines the production rate of bioink; thus, the precise control of curing is critical [100]. Thus, the large scale production of effective hydrogels in complex requires skilled chemical understanding and well-documented, strict regulations.

#### 4.2.3. Long-Term Stability

Degradation of hydrogels is another sensitive issue that must be addressed to harness the full applicability of hydrogels. As more and more energy sectors demand improved hydrogels to perform complex functions, their stable operation is essential [101]. This degradation can become even more critical when hydrogels are applied in life-saving fields such as biomedical applications; thus, developing simple strategies to control the degradation of hydrogels is immediately needed.

One of the possible and effective approaches to effectively control the degradation in the hydrogel is to precisely tune the rate of dissociation of cross-linked polymer chains. Recent studies revealed that controlling the various size mismatches between the segments can be a handy tool to regulate and modulate the dissociation rate of polymers [101]. Modifying the crosslinking density within the hydrogel network is another effective strategy to control hydrogel degradation. Increasing crosslinking density typically leads to slower degradation [102]. This can be achieved by adjusting the concentration of crosslinking agents during synthesis. In addition, altering the structure of the hydrogel, for instance, introducing interpenetrating networks or incorporating nanoparticles, can affect degradation rates—the process commonly known as the additive effect [103]. Fine-tuning the hydrogel architecture can provide control over its susceptibility to degradation. Hydrogels are pH sensitive and thus degrade quickly when the external environment changes; however, designing hydrogels with temperature sensitivity may offer some degradation control [104].

#### 4.2.4. Cost Consideration

The cost consideration of any developing technology is an essential step in forecasting its market viability, and hydrogel is not an exception. The production cost is a critical step that becomes significant when mass production of material is to be considered. The same can be seen in other techniques where research has been shifting to more economical precursors. Consider the shift of energy researchers to sodium-ion batteries from LIB technology [6,105], the halide perovskite shift from Si-based solar cell technology [106], and so on. Primarily, the raw materials for hydrogel synthesis are sometimes very costly, ultimately increasing their overall price. On the other hand, hydrogel production costs also depend on the synthesis methods, scale of production, and customization. The choice of polymers and crosslinking agents, as well as the need for post-processing steps and regulatory compliance, all contribute to the overall expense. Additionally, the demand, competition, and specific applications of hydrogels can influence pricing. Balancing these considerations is crucial for assessing the cost-effectiveness of hydrogels, particularly in industries like healthcare, where biocompatibility and regulatory standards play a significant role. Advances in manufacturing technologies and economies of scale may contribute to reducing costs over time.

#### 4.2.5. Material Selection

The choice of active materials for the complex designing of hydrogels required in intricate biomedical applications is a need of the day. Some hydrogel precursors rely on resources that are not plentiful and may have severe environmental concerns. Water pollution by releasing heavy metals is a significant environmental concern for hydrogels [107]. Ensuring a sustainable source for the precursor is a wise idea and must be strived for if hydrogels are seen as an alternative to a fossil-fuel-based economy. In the realm of hydrogels, material selection is pivotal as it directly affects their ability to absorb and retain water, biocompatibility, and overall performance in applications like drug delivery, tissue engineering, and sensors. Engineers and researchers consider a range of natural and synthetic polymers, crosslinkers, and additives when designing hydrogels, ensuring that the selected materials align with the intended purpose and provide optimal performance for the given conditions. The judicious choice of materials is essential for achieving desired outcomes and addressing challenges related to the specific application of hydrogels.

#### 4.2.6. Regulations and Environmental Concerns

An extension of the material selection, it is essential to understand that there must be strict regulation on the extraction of precursor extractions to device fabrication, particularly in biomedical applications. Though the use of hydrogel has offered an alternative against conventional fuels for the modern electrified society, several issues must be addressed. However, it is crucial to consider potential challenges, such as the environmental impact of hydrogel production and end-of-life considerations. Rigorous life cycle assessments and responsible waste management practices are essential to ensure the overall sustainability of hydrogel-based energy storage systems. The effectiveness of hydrogels can be estimated through a recent study that reveals ~one tonne of acrylamide hydrogel (AH) costs $2727 to import, compared to $1106 for one tonne of straw-based hydrogel (SH) [108]. The economic aspect is actualized not only through affordability but also by employing agricultural residues like rice straw. These materials are sustainable, renewable, and biodegradable, further enhancing the economic dimension.

#### 4.2.7. Integration with Existing Technologies

Integrating hydrogel-based energy storage with existing technologies poses various challenges that must be carefully addressed. Compatibility issues with the current energy infrastructure, including power grids and storage systems, must be thoroughly examined. Standardization efforts become crucial to ensure uniformity and compatibility across different energy storage technologies. Collaborative initiatives aimed at establishing industry-wide standards can play a pivotal role in streamlining integration processes. Potential solutions may involve developing adaptable interfaces, protocols, and communication systems that facilitate the seamless incorporation of hydrogel-based energy storage into the existing energy landscape. Addressing these integration challenges is vital for successfully adopting hydrogel technologies, contributing to a more resilient and efficient energy infrastructure.

#### 4.2.8. Lab to Field

The transition from the laboratory to real-world field applications represents a pivotal stage for hydrogels, showcasing their transformative potential. In the controlled environment of the lab, researchers meticulously design and optimize hydrogel formulations, tailoring their properties for specific functions like drug delivery, wound healing, or environmental remediation. However, the true impact of hydrogels is realized when they make the journey from the lab to practical field applications. This shift involves addressing challenges related to scalability, reproducibility, and adaptability to diverse and dynamic real-world conditions. As hydrogels venture into the field, their effectiveness in addressing complex challenges, whether in medical treatments or environmental solutions, becomes evident. This journey exemplifies the bridge between scientific innovation and tangible societal benefits, showcasing the versatility and promise of hydrogels in making a meaningful impact on a broader scale.

## 5. Future Prospects and Summary

Despite having reached a satisfactory stage in energy storage devices to foresee achievable carbon neutrality and net zero emissions, protecting our world from global warming due to burning fossil fuels is a daunting task ahead [109]. As the pursuit of green energy generation and effective storage gains momentum, researchers are making significant strides, although full substitution of fossil fuels remains a distant goal. The sun and water, being the sole sources of green energy devoid of carbonaceous gas emissions, drive interest primarily towards photovoltaics and fuel cells, with supplementary attention to wind, tidal, and hydro energy. Beyond energy generation, the storage aspect, accomplished through batteries and supercapacitors leveraging chemical processes, marks crucial advancements. The need for energy extends beyond powering heavy vehicles, reaching into the realms of computers, mobile phones, and especially biology/biotechnology.

Consequently, energy generation and storage devices necessitate diverse attributes like varying shapes, sizes, flexibility, self-healing, elasticity, and biocompatibility. Addressing these requirements, polymer gels emerge as pivotal materials. Therefore, this review focuses on the application of polymer gels as key components in energy materials, exploring their potential in shaping the future of energy technologies [110,111,112].

Hence, it is evident from the preceding discussions that hybrid conducting polymer gels play a crucial role in constructing various energy devices, including fuel cells, batteries, and supercapacitors. Notably, the discussion here does not encompass the significant role of polymer gels in bioenergy, which pertains to energy associated with biological processes. Considering its relative neglect, this area presents a substantial scope for future exploration and holds promise for further research and development. 

Gels exhibit significant promise in advancing energy storage technologies, particularly in supercapacitors, fuel cells, and batteries. In supercapacitors, gel electrolytes enhance safety and conductivity, improving performance. With their unique properties, gel polymer electrolytes play a vital role in fuel cells, offering stability and efficient ion transport. Also, gels like those formed in hybrid systems enhance electrode stability and overall battery performance. The use of hydrogel can bring wonders to energy storage technology. Their unique properties, such as high water content, tunable mechanical strength, and biocompatibility, make them ideal candidates for applications like drug delivery, wound healing, tissue engineering, and environmental remediation. Hydrogels can mimic the properties of natural tissues, providing a supportive environment for cell growth and regeneration. They exhibit responsiveness to environmental stimuli, such as pH and temperature, enabling controlled drug release and other dynamic functionalities. In the field of energy storage, hydrogels hold promise for creating flexible and sustainable solutions. However, challenges such as degradation control, material selection, and large-scale production need to be addressed for optimal utilization. As hydrogels continue to bridge the gap from laboratory research to real-world applications, their potential to revolutionize various industries and address pressing global challenges becomes increasingly apparent. The prospects of gels in these energy storage applications involve ongoing research to optimize their formulations, tailor their properties, and explore novel materials. The versatile nature of gels holds the potential to address challenges in energy storage and propel advancements in sustainable and efficient technologies.

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
