# Peer review of "Gels in Motion: Recent Advancements in Energy Applications"

_gels, 2024, doi:10.3390/gels10020122_

Round 1

Reviewer 1 Report

Comments and Suggestions for Authors

This review focuses on the promising applications of gels, particularly hydrogels, in advancing next-gen batteries and supercapacitors for energy storage. It highlights the strength, flexibility, and charge transport capabilities of hydrogels, along with progress in creating stimuli-responsive and self-healing hydrogels.

The manuscript showcases versatile uses of hydrogels in energy storage, including tailorable architectures, conductive nanostructures, 3D frameworks, and multifunctionalities. While suitable for the journal, it needs expansion in specific areas for better clarity.

    • Elaborate on hydrogels' chemical and physical properties, including structure, composition, and behavior in different environments.
    • Explain the principles behind stimuli-responsive and self-healing characteristics, delving into underlying chemical and physical mechanisms.
    • Discuss the electrochemical processes in energy storage technologies and how hydrogels interact with them.
    • Address challenges in scaling up hydrogel-based energy storage, covering manufacturing, cost, and long-term stability.
    • Explore limitations of current hydrogel-based devices, such as energy density, cycle life, and environmental impact. Suggest potential solutions or areas for future research.
    • Analyze economic and environmental impact if hydrogel-based energy storage is widely adopted. Consider their role in transitioning to a more sustainable energy infrastructure.
    • Discuss the implications of using hydrogels in energy storage on global energy consumption, greenhouse gas emissions, and resource utilization. Provide a comprehensive overview of potential benefits and challenges.

Incorporating these improvements will enhance the manuscript's exploration of hydrogels in energy storage technologies, making it more comprehensive and engaging to the readers.

Reviewer 2 Report

Comments and Suggestions for Authors

The manuscript is eligible for acceptance following minor corrections

1.     The manuscript discusses polymer gel applications in batteries, supercapacitors, and fuel cells. The term 'gel' in the title appears too general and needs improvement to accurately reflect the manuscript's content.

2.     Please include fuel cells in the abstract.

3.     I am uncertain about the details regarding polymer gel electrodes. It would be helpful to include more information about the properties of the electrode materials and an explanation of why they are classified as gel types in this section

Reviewer 3 Report

Comments and Suggestions for Authors

The manuscript is a short review of the recent progress of gels for energy and storage technologies. The topic is very interesting but for its publication I think that the authors should expand the information. Please, add more information for each technology (super-capacitors, fuel cells, batteries, etc.) that you include in your manuscript.

Comments on the Quality of English Language

The authors shoud revised English (minor editing).
